# Development and Validation of a Parental Health-Related Empowerment Scale with Low Income Parents

**DOI:** 10.3390/ijerph17228645

**Published:** 2020-11-20

**Authors:** Roger Figueroa, Cristina M. Gago, Jacob Beckerman-Hsu, Alyssa Aftosmes-Tobio, Xinting Yu, Kirsten K. Davison, Janine J. Jurkowski

**Affiliations:** 1Division of Nutritional Sciences, College of Human Ecology, Cornell University, Ithaca, NY 14853, USA; 2Department of Nutrition, Harvard T.H. Chan School of Public Health, Harvard University, Boston, MA 02115, USA; gago@g.harvard.edu (C.M.G.); xinting.yu@bc.edu (X.Y.); 3School of Social Work, Boston College, Chestnut Hill, MA 02467, USA; jacob.beckerman@bc.edu (J.B.-H.); aftosmes@bc.edu (A.A.-T.); kirsten.davison@bc.edu (K.K.D.); 4Department of Health Policy, Management, and Behavior, University at Albany School of Public Health, State University of New York, Albany, NY 12144, USA; jjurkowski@albany.edu

**Keywords:** factorial validity, empowerment theory, head start, intervention, parental empowerment, scale development

## Abstract

Objectives: Consistent with empowerment theory, parental empowerment acts as a mechanism of change in family-based interventions to support child health. Yet, there are no comprehensive, validated measures of parental health-related empowerment to test this important perspective. Informed by empowerment theory and in the context of a community-based obesity intervention, we developed a self-report measure of parental health-related empowerment and tested its preliminary validity with low-income parents. Methods: The Parental Empowerment through Awareness, Relationships, and Resources (PEARR) is a 21-item scale designed to measure three subdimensions of empowerment including resource empowerment, critical awareness, and relational empowerment. In the fall of 2017 or the fall of 2018, low-income parents (n = 770, 88% mothers) from 16 Head Start programs in Greater Boston completed the PEARR. The resulting data were randomly split into two equal samples with complete data. The factorial structure of the PEARR was tested in the first half of the sample using principal component analysis (PCA) and exploratory factor analysis (EFA) and subsequently confirmed with the second half of the sample using confirmatory factor analysis (CFA). Internal consistency coefficients were calculated for the final subscales. Results: Results from the PCA and EFA analyses identified three component factors (eigenvalues = 8.25, 2.75, 2.12) with all items loading significantly onto the hypothesized subdimension (β > 0.59 and *p* < 0.01). The three-factor model was subsequently confirmed with the second half of the sample using CFA (β > 0.54 and *p* < 0.01). Fit indices met minimum criteria (Comparative Fit Index = 0.95, Root Mean Square Error of Approximation = 0.05 (0.05, 0.06), Standardized Root-Mean-Square Residual = 0.05). Subscales demonstrated strong internal consistency (α= 0.83–0.90). Conclusions: Results support initial validity of a brief survey measuring parental empowerment for child health among Head Start parents. The PEARR can be utilized to measure changes in parental empowerment through interventions targeting empowerment as a mechanism of change.

## 1. Introduction

Income inequalities in the United States (U.S.) are at the highest level in 50 years [1]. U.S. adults and children from low-income backgrounds experience a greater burden of health disparities than their counterparts with higher socioeconomic status (SES) that persist across many health indicators [2,3]. Low-income populations typically lack access to power structures, and therefore lack resources, networks, and capacity to overcome a myriad of social and environmental barriers to maintaining health and wellness [4,5]. The Communities for Healthy Living (CHL) program aims to empower parents to promote child health and prevent childhood obesity through parental empowerment [6,7,8]. The CHL’s innovative intervention leverages child health resource channels, is delivered in partnership with Head Start, and specifically targets three psychological empowerment constructs, i.e., critical awareness, resource empowerment, and relational empowerment, through its 10-week parent-led wellness education curriculum and complementary components [6,7,8].

Broadly, psychological empowerment is the process by which people gain greater control over their lives, participate in democratic decision making, and develop critical awareness of their sociopolitical environments [4,5]. Within the multilevel umbrella of empowerment theory, Zimmerman defined psychological empowerment as consisting of individual-level emotional (i.e., intrapersonal), cognitive (i.e., interactional), and behavioral components (i.e., actions taken) [5,9,10,11]. As a facet of cognitive psychological empowerment, critical awareness is defined as an in-depth understanding of one’s life situation and its contributing factors [11,12]. This interactional empowerment requires individuals to identify causal agents in their environments, and furthermore, to engage with them to accomplish a given goal [11,12]. Critical awareness also implicates environmental mastery, which is defined as the ability to choose or change the surrounding context through action [11].

Serving as a second facet of cognitive psychological empowerment, resource empowerment, also known as resource mobilization, is defined as the ability to identify and gain access to health-enhancing resources, which involves mastery of skills for obtaining resources to promote family health and well-being [5,9,10,11,12]. Resource empowerment focuses on an individual’s awareness of and engagement with resources [9].

As an extension of the model by Zimmerman’s, Christens posited a third psychological empowerment arm, relational empowerment (i.e., interpersonal), which he defined as the awareness of and ability to utilize social relationships to improve one’s life situation [9,13,14,15]. This expansion on the traditional psychological empowerment individualism serves to emphasize the significance of actualizing control, in addition to feeling in control [9]. Furthermore, it serves to capture the multidimensional nature of psychological empowerment by highlighting the significance of interpersonal relationships in psychosocial dynamics, independent of the extra-individual context [9,15,16,17].

Within the context of health promotion, the World Health Organization and academic researchers have repeatedly identified empowerment as a central concept, required for the successful closure of multilevel disparities affecting marginalized communities [18,19,20]. Consequently, psychological empowerment has been used to inform numerous community-based health education programs and behavior change interventions over the past four decades [21,22]. Offering a high degree of adaptability as a framework, psychological empowerment has informed a diverse range of program behavior targets, including that of tobacco cessation, HIV prevention, and substance abuse control [23,24,25]. Furthermore, as a framework for intervention development, psychological empowerment has been shown to fit programming for both children and adult audiences [26,27]. Researchers hypothesize that this consistently observed connection between empowerment and behavior change may be driven by the underlying engagement of community participation and a heightened sense of community offered by intervention contexts [28,29,30,31].

While psychological empowerment constructs have been extensively applied and studied within the contexts of health services and community contexts [31,32,33,34,35,36,37], few valid health-related empowerment scales exist [31,34,37,38,39]. Among those that do, even fewer have undergone thorough validity and reliability testing. Systematic review evidence has suggested that although health-related empowerment scales have been tested, constructs such as critical awareness, as well as resource and relational empowerment have not been tested and validated for low-income parents [38]. The current study is the first to measure psychological empowerment constructs within the setting of a family-centered, community-based obesity prevention program [6,8,40]. Furthermore, this is the first evaluation of the reliability and validity of these key psychological empowerment constructs, i.e., resource empowerment, critical awareness, and relational empowerment, amongst low-income parents of preschoolers.

## 2. Materials and Methods

### 2.1. Study Sample

This study used cross-sectional data from a clinical trial based in the United States. The ethics boards of Harvard T. H. Chan School of Public Health (#IRB15-3559, 18 November 2015) and Boston College (#20.005.01, 19 June 2019) reviewed and approved the research conducted. The main focus of the Communities for Healthy Living (CHL) clinical trial is to prevent childhood obesity among low-income families with preschoolers using a community-based participatory research (CBPR) approach [6]. Empowerment is a hypothesized mechanism of change in this 3-year trial, and it is measured each year in a convenience sample of parents.

One parent or primary caregiver per family, with a child aged 3–5 years enrolled in one of 16 Head Start programs in the greater Boston area, was recruited to complete a 30-min survey between September and November 2017 and October and December 2018. Approximately 30% of the parents were recruited from each Head Start program to ensure the resulting sample was representative of all families enrolled in the 16 programs. For the current study, parents’ responses to the empowerment measures, from the fall of 2017 to the 2018 administrations of the survey (<770), were combined with demographic data from administrative records. Parents provided consent to link their survey responses with their extracted demographic data (e.g., race/ethnicity, parent education). Upon providing informed consent, parents completed the survey, in Chinese (n = 15), English (n = 611), or Spanish (n = 181). These versions were translated and back translated by members of the research team that were native language speakers. Parents received support from a bilingual research assistant as needed. This study focused on the items measuring parental empowerment including resource empowerment, critical awareness of socio-ecological influences of health, and relational empowerment constructs. 

### 2.2. Measures

#### 2.2.1. Parental Empowerment

##### Item Development 

The hypothesized psychological empowerment dimensions in the current study were informed by prior formative research [8]. During the formative research stage, a preliminary pool of 70 items spanning a wide range of empowerment concepts (e.g., collaborative competence, bridging social divides, facilitating others, relational empowerment, mobilizing networks, critical consciousness) was developed. Following an extensive review of the literature, the authors elected to prioritize the constructs of critical consciousness, and relational and resource empowerment [9,11,12,13,14,15,37]. Using an expert agreement process, four of the authors (R.F., J.B., K.K.D. and J.J.J.) developed a preliminary version of the scale with 26 items. Following pilot testing (described below) 5 items were removed resulting in a final scale with 21 items.

Items measuring resource empowerment (n = 9), defined as the ability to identify and gain access to health-enhancing resources, were drawn from existing literature on resource empowerment [9,13,14,37]. The items measured two aspects of resource empowerment, including knowledge and skill development across life (e.g., I know “who can,” “what can,” and “how to” help my family, 5 items) and practices to access resources (e.g., I “ask” and “use” resources in my community, 4 items). These domains were selected given their role as obesity-related correlates and as a key empowerment target in the larger clinical trial [6,8,40,41,42].

Items assessing critical awareness of health (n = 7), defined as parent awareness of factors influencing health across ecological levels, were adapted from the critical consciousness scale [11,12]. Items assessed factors at the level of the family (i.e., parents’ health behaviors), household (e.g., type of housing), and community (e.g., neighborhood, Head Start policies).

Lastly, relational empowerment, defined as an awareness of and ability to utilize social relationships to improve one’s life situation, was measured using items (n = 5) drawn from the existing literature on relational empowerment [9,13,14,15]. Items measured parents’ ability to share and communicate with others regarding shared challenges and resources (i.e., “I share what I know or learn about health with other parents”, “I talk to other parents about my problems”, and “I talk to other parents to get information and resources for my family”). Across all subscales, participants were instructed to report how much they agreed or disagreed with the statements in each scale. All response options for each item ranged from strongly disagree (1) to strongly agree (4) on a 4-point scale.

##### Pilot Testing Items

A sample of 25 Head Start parents completed the preliminary 26 item version of the survey at the formative stages of the larger clinical trial (2016–2017). Participants completed the full survey, and subsequently, some respondents also answered quality improvement questions (n = 18), while others completed the full survey with cognitive interview questions built in (n = 7). Following the results from the analyses of these 25 cases, empowerment-related indicators were refined to improve their clarity and 5 poorly performing items were deleted. This study only included indicators that were revised and retained following the pilot-test stage.

#### 2.2.2. Demographic Factors

Information on family demographic factors (respondent age, sex, education, race/ethnicity, and birthplace) was extracted from Head Start records; parents gave passive consent to access administrative data, which was linked to parents’ responses on the empowerment scale.

### 2.3. Statistical Analysis

Prior to data analysis, the sample was randomly divided into two even groups. The validity of the parental empowerment through awareness, relationships, and resources (PEARR) scale was then tested in two stages. Analyses were conducted in Stata (StataCorp, College Station, TX, USA), version 14 between July 2019 and February 2020.

Using the full sample, polychoric correlation was used to assess associations across indicators in each subscale. Stage 1 utilized data from the first half of the randomly divided sample and examined the scale’s preliminary factor structure. First, principal component analysis (PCA) was performed to determine the number of factors to retain (i.e., the number of eigenvalues greater than 1). The factor loadings were also examined at this stage to identify, and if necessary, remove, low performing items (i.e., with a factor loading <0.40). Next, EFA was performed, specifying the number of factors identified in the PCA, to assess whether the items denoting each hypothesized parental empowerment factor loaded on to the expected factor. Finally, factor loads were rotated to get a clearer pattern of the items denoting each hypothesized parental empowerment (i.e., final factor solution).

Using data from the second randomly split subsample, Stage 2 tested and confirmed the factor structure identified in Stage 1. The hypothesized factor structure was tested using confirmatory factor analysis (CFA). Model fit was assessed using the root mean square error of approximation (RMSEA), standardized root-mean-square residual (SRMR), and the comparative fit index (CFI) [43]. In order for our models to attain acceptable model fit, at least two of the following model fit indicators had to meet the following criteria: RMSEA (≤0.08), SRMR (≤0.10), and CFI (≥0.90). At the conclusion of the Stage 2 analyses, empowerment subscales were created (with each subscale being the average of the items loading onto that factor) and summary statistics were calculated including internal consistency coefficients (i.e., Cronbach’s α), cross factor correlations, as well as correlations with validated subscales on obesity-related parenting practices (i.e., concurrent validity) [44].

## 3. Results

Participants (n = 770) included 679 (88%) mothers and 91 (12%) fathers. Parents on average were 34 years of age and over half (60%) had a high school education or less. The majority of parents identified racially as Black or African American or ethnically as Hispanic/Latino (Table 1).

The results from the polychoric correlation matrix indicated that the items correlated significantly within each hypothesized psychological empowerment dimension (r = 0.38–0.87). In Stage 1, three component factors were identified in the PCA, with eigenvalues of 8.25, 2.75, and 2.12. Subsequently, results from the EFA yielded a final three-factor solution with eigenvalues of 5.12, 4.81, and 3.19. The EFA model supported the hypothesized three-factor model for resource empowerment (nine items), critical awareness (seven items), and relational empowerment (five items) as a preliminary factor structure (Table 2). All items had factor loadings greater than 0.40 (*β* > 0.59) following EFA, and therefore were retained. Individually, indicators in the resource empowerment dimension explained 24.40% of its variance, indicators in the critical awareness dimension explained 22.94% of its variance, and indicators in the relational empowerment dimension explained 15.20% of its variance (62.54% overall variance explained across domains).

In Stage 2, CFA results confirmed the three-factor model identified in Stage 1 (*β* > 0.54 and *p* < 0.01). The final factor structure, factor loadings, and model fit statistics for each parental empowerment dimension of the PEARR are listed in Table 3. Fit indices met minimum criteria (*X*^2^ (147, *n* = 337) = 373.78, *p* < 0.01; Comparative Fit Index = 0.95; Root Mean Square Error of Approximation = 0.05 (0.05, 0.06); Standardized Root-Mean-Square Residual = 0.05). Cross factor correlations were significantly moderate (r = 0.38–0.40). Items showed strong internal consistency across each of their respective dimensions (α = 0.83–0.90). Lastly, the PEARR scale showed modest concurrent validity in relation to obesity-related parenting practices focused on physical activity (r = 0.25–0.29) and sleep (r = 0.25–0.37), but not food-related parenting practices (r = 0.02–0.16). Summary statistics for each item in the final factor structure for each parental empowerment dimension can be found in Table 4.

## 4. Discussion

The process of validating psychological constructs through scale development and psychometric testing varies. At its core, it requires precise and detailed conceptualization of the target construct, theoretical context, and systematic testing of the cohesion across variables observed denoting hypothesized dimensions. Our research team developed a measure of health-related parental empowerment using theoretical constructs from empowerment-relevant theories and assessed its factorial validity. The PEARR scale, a brief health-related parental empowerment measure, was developed and was found to demonstrate substantial factorial validity and strong internal consistency across dimensions.

There is limited research thoroughly assessing the validity and reliability of health-related psychological empowerment within the contexts of health services and community-based interventions. The systematic review evidence suggested that multiple studies have made significant contributions to the field within the context of their study subject, which influenced our efforts [38]. However, the main content areas and target audiences for which health-related psychological empowerment scales have been developed do not focus on low-income parents of typically developing children, and do not assess critical awareness, resource and relational empowerment. The systematic review evidence also suggested that individual-level psychological empowerment was heavily prioritized in validation studies over community- or organizational-level psychological empowerment. Most studies validating individual-level empowerment measured at least one approach or a combination of content, structural, internal, and external construct validity methods [34,37,38,39].

This study evaluates the coherence, consistency, and dimensionality of a parental empowerment scale specific to child health among a diverse sample of Head Start parents and offers a preliminary basis of its validity and psychometric properties. Through a series of systematic steps, we find factorial validity across three psychological empowerment constructs through 21 observed indicators. As such, this scale may be a useful instrument within the context of community programming to assess critical gaps in empowerment within certain programs, or it could be used to evaluate programs and interventions which specifically target psychological empowerment. This scale could be a useful tool for studies within the context of health promotion (i.e., chronic disease and obesity prevention) for caregivers in general, although less applicable within elderly care contexts. The research team also can see adapted versions of the PEARR scale expand its applicability to additional research contexts. For instance, slight adjustments to the wording could make the current version of the scale more relevant to other health topics or populations. Indicators referencing Head Start could be changed to fit the scope of a different program and administered as most relevant to the study context, although in such a case, the research team recommends assessing the sensitivity of the adapted items with remaining items in the scale.

This new measure was designed for an intervention that focuses on empowerment as a mechanism for health behavior change to prevent childhood obesity. In the future, the research team could pursue strategies to further validate the scales (i.e., invariance across multiple groups, scale efficiency assessment, and test-retest reliability).

Because psychological empowerment is a complex multilevel construct, findings from our study may not capture other relevant aspects of psychological empowerment (i.e., community, organizational). In addition, although our study makes efforts to represent a significant subsample of diverse parent audiences across demographic characteristics (e.g., parent gender and ethnicity), future studies should further test whether the measure is invariant across demographic factors, such as gender and primary language spoken. For instance, addressing the imbalance of mothers and fathers may minimize bias from the majority representing the parent sample.

## 5. Conclusions

Our study contributes to the larger body of literature by analyzing a relatively large sample of Head Start parents to assess psychological empowerment using a brief instrument. We believe this instrument is one of the first valid assessments of parental psychological empowerment within the context of early childhood education for promoting health. Future research should examine the validity of the factor structure across language and parent gender along with the predictive validity of the scale in reference to behavioral change.

## Figures and Tables

**Table 1 ijerph-17-08645-t001:** Demographic characteristics of study participants (n = 770).

	Summary Statistic
Age (M, SD)	34.13 (±6.95)
Sex (*n*, %)	
Male	91 (11.82%)
Female	679 (88.18%)
Education (*n*, %)	
HS degree or less	463 (60.12%)
Some college/associates degree	180 (23.37%)
4-year college degree and above	92 (11.94%)
Other	35 (4.54%)
Race (*n*, %)	
Asian	70 (9.09%)
Black	304 (39.48%)
White/Caucasian	86 (11.16%)
Other	292 (37.92%)
Ethnicity (*n*, %)	
Hispanic/Latino	294 (38.18%)

Note: Due to missing data, some categories may not sum to 100% of the study sample. The category “other” includes parents who identified as biracial/multiracial, as well as descendant of various Latin-American countries (i.e., Brazil, Colombia, Dominican Republic, Guatemala, Haiti, El Salvador, Puerto Rico), China, Ethiopia, Morocco, Somalia, among others.

**Table 2 ijerph-17-08645-t002:** Exploratory factor analysis results for the psychological empowerment scale with first randomly split subsample (n = 339).

	Factor Loadings		
Item	Resource Empowerment	Critical Awareness of SE Influences of Health	Relational Empowerment
1. I know who to speak with to help my child.	0.77	0.19	−0.02
2. I know what questions to ask to help my child.	0.81	0.10	0.01
3. I know I can get my family to help.	0.79	0.03	0.08
4. I know how to find programs, services, or other resources in my community.	0.67	0.03	0.14
5. I know how to speak up or advocate for my child with professionals.	0.78	0.18	0.05
6. I ask my child’s doctor for help or advice.	0.75	0.22	0.14
7. I ask friends and family for help or advice.	0.70	0.11	0.28
8. I ask a Head Start teacher or staff for help or advice.	0.62	0.17	0.21
9. I use the programs, services, or other resources in my community to help my child.	0.59	0.17	0.24
10. Parents’ health behaviors (for example, nutrition, physical activity, sleep) influence children’s health.	0.20	0.80	0.07
11. Parenting practices (for example, rules around bedtime) influence children’s health.	0.16	0.79	0.12
12. Behaviors of family members influence children’s health.	0.11	0.84	0.15
13. The house, apartment or structure families live in influence children’s health.	0.11	0.85	0.12
14. The neighborhood that families live in influence children’s health.	0.06	0.75	0.19
15. The things children see on television or in electronic games influence their health.	0.01	0.68	0.14
16. Head Start influences children’s health.	0.17	0.68	0.17
17. I share what I know or learn about health with other parents.	0.15	0.29	0.64
18. I share what I know or learn about health on social media, such as Facebook or Instagram, with other parents.	0.01	0.15	0.63
19. I can rely on other parents for help when I need it.	0.09	0.13	0.80
20. I can talk about my problems with other parents.	0.08	0.13	0.83
21. I talk to other parents to get information and resources for my family.	0.13	0.18	0.80
Eigenvalues	5.12	4.81	3.19
% of variance	24.40	22.94	15.20

**Table 3 ijerph-17-08645-t003:** Confirmatory factor analysis results for the final 3-factor model of the psychological empowerment scale with second randomly split subsample (n = 337, *p* < 0.01).

	Factor Loadings		
Item	Resource Empowerment	Critical Awareness of SE Influences of Health	Relational Empowerment
1. I know who to speak with to help my child.	0.62		
2. I know what questions to ask to help my child.	0.71		
3. I know I can get my family to help.	0.68		
4. I know how to find programs, services, or other resources in my community.	0.67		
5. I know how to speak up or advocate for my child with professionals.	0.77		
6. I ask my child’s doctor for help or advice.	0.72		
7. I ask friends and family for help or advice.	0.65		
8. I ask a Head Start teacher or staff for help or advice.	0.54		
9. I use the programs, services, or other resources in my community to help my child.	0.66		
10. Parents’ health behaviors (for example, nutrition, physical activity, sleep) influence children’s health.		0.75	
11. Parenting practices (for example, rules around bedtime) influence children’s health.		0.80	
12. Behaviors of family members influence children’s health.		0.79	
13. The house, apartment or structure families live in influence children’s health.		0.80	
14. The neighborhood that families live in influence children’s health.		0.67	
15. The things children see on television or in electronic games influence their health.		0.64	
16. Head Start influences children’s health.		0.73	
17. I share what I know or learn about health with other parents.			0.70
18. I share what I know or learn about health on social media, such as Facebook or Instagram, with other parents.			0.61
19. I can rely on other parents for help when I need it.			0.64
20. I can talk about my problems with other parents.			0.64
21. I talk to other parents to get information and resources for my family.			0.75
Model fit statistics: *X*^2^ (147, *n* = 337) = 373.78, *p* < 0.01; CFI = 0.95; RMSEA = 0.05 (0.05, 0.06); SRMR = 0.05.			

**Table 4 ijerph-17-08645-t004:** Summary statistics and internal consistency coefficients of the parental empowerment scale separately for each subscale in the total sample.

Subscales and Associated Items	Mean (SD)
Resource Empowerment	
1. I know who to speak with to help my child	3.37 (0.64)
2. I know what questions to ask to help my child	3.38 (0.58)
3. I know I can get my family to help	3.30 (0.67)
4. I know how to find programs, services, or other resources in my community	3.13 (0.74)
5. I know how to speak up or advocate for my child with professionals	3.31 (0.64)
6. I ask my child’s doctor for help or advice	3.37 (0.63)
7. I ask friends and family for help or advice	3.22 (0.67)
8. I ask a Head Start teacher or staff for help or advice	3.19 (0.69)
9. I use the programs, services, or other resources in my community to help my child	3.13 (0.70)
Overall Resource Empowerment, mean (SD)	3.28 (0.48)
Critical Awareness of Socio-Ecological Influences of Health	
10. Parents’ health behaviors (for example, nutrition, physical activity, sleep) influence children’s health	3.42 (0.62)
11. Parenting practices (for example, rules around bedtime) influence children’s health	3.42 (0.62)
12. Behaviors of family members influence children’s health	3.28 (0.70)
13. The house, apartment or structure families live in influence children’s health	3.31 (0.71)
14. The neighborhood that families live in influence children’s health	3.14 (0.79)
15. The things children see on television or in electronic games influence their health	3.21 (0.78)
16. Head Start influences children’s health	3.37 (0.68)
Overall Critical Awareness of SE Influences of Health, mean (SD)	3.32 (0.56)
Relational Empowerment	
17. I share what I know or learn about health with other parents	3.05 (0.66)
18. I share what I know or learn about health on social media, such as Facebook or Instagram, with other parents	2.70 (0.80)
19. I can rely on other parents for help when I need it	2.76 (0.75)
20. I can talk about my problems with other parents.	2.69 (0.76)
21. I talk to other parents to get information and resources for my family	2.88 (0.70)
Overall Relational Empowerment, mean (SD)	2.82 (0.57)

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
