# Peer review of "Development and Validation of a Parental Health-Related Empowerment Scale with Low Income Parents"

_ijerph, 2020, doi:10.3390/ijerph17228645_

Round 1
Reviewer 1 Report
The authors have developed and validated a very important scale to measure health-related empowerment. Overall, the paper reads quite well. I would give some recommendations for further improvement. 1. As there were three language versions of this scale, did the authors pay attention to the translational equivalence? (e.g., back translation). If possible, please also check measurement invariance across different ethnic groups or language groups. 2. There are much more female respondents than the male respondents. This imbalance may create bias. I would suggest the author to check the measurement invariance across male and female groups.Author Response
|
Reviewer 1 Comments |
Author’s response: |
Corrections & Locations |
|
The authors have developed and validated a very important scale to measure health-related empowerment. Overall, the paper reads quite well. I would give some recommendations for further improvement. |
Thank you for your feedback. We field your recommendation below. |
N/A |
|
1. As there were three language versions of this scale, did the authors pay attention to the translational equivalence? (e.g., back translation). If possible, please also check measurement invariance across different ethnic groups or language groups. |
Thank you for your question. Each version of the scale was translated and back translated by native language speakers from the original English version. The process was also supported by members of the research team that were fluent in each language. We added a sentence clarifying this step in the manuscript. |
Pg. 3, Ln. 110-111
“These versions were translated and back translated by members of the research team that were native language speakers.” |
|
|
RE: measurement Invariance - We are doing additional validity and reliability work in a forthcoming paper that includes analysis of measure invariance by language. Therefore, it is not included in this particular manuscript. |
|
|
2. There are much more female respondents than the male respondents. This imbalance may create bias. I would suggest the author to check the measurement invariance across male and female groups. |
Thank you for this observation. We are aware of this potential limitation. It is quite rare to recruit as many fathers as we did. As mentioned in our previous response, we are working on a forthcoming manuscript that will assess measure invariance by language and parent gender. |
Pg. 6, Ln. 471-472
“For instance, addressing the imbalance of mothers and fathers may minimize bias from the majority representing the parent sample.” |
|
|
We added a statement in the limitation section to address your |
|
Reviewer 2 Report
The paper “Development and validation of a parent health-related 1 empowerment scale with low-income parents” reports the development of a 21-item scale to measure parental empowerment through awareness, relationships, and resources (PEARR).
Please include the following information.
- Indicate how many items were part of the initial pool of items, if the initial pool was different from the final scale.
- Indicate how many items were developed and how many were took from previous scales.
- How the process of construct validity was conducted? e.g., expert agreement.
- Include details about PCA and EFA, e.g., rotation.
- Report the range and distribution of items responses.
- Indicate why a PCA was conducted instead of other procedures/criteria to estimate the initial number of factors.
- Report chi-scare and degree of freedom for the 3-factor model.
- Report the scale’ instructions
- Report ordinal omega if the Likert scale has less than 10 points.
- Evidence concurrent/discriminant validity. How this scale is correlated with other measures.
Please clarify
The correlation coefficient between the 3-factor was obtained using each scale item's added scores or was obtained directly from the CFA model.
- The response to the items is a frequency/agreement Likert scale? How many points the scale has?
- Were the analyses conducted considering the data as ordinal data? Were the polychromic matrix of correlation computed? Please indicate the algorithm used to obtain the factors for CFA and EFA.
- Items 8 and 16 (I ask a Head Start teacher or staff for help or advice; Head Start influences children’s health) are specific to an intervention or program. Please discuss the role of these items in a study outside of the Head Start program. The items should be changed? Should be eliminated if the scale is administered in a different program?
- How the items from the scale “Critical Awareness of Socio-Ecological Influences of Health” can be differentiated from beliefs or cultural biases, for example, the item “The things children see on television or in electronic games influence their health.”
Please discuss in the limitation section
- The impact of the gender bias of the sample
Author Response
|
|
concern. |
|
|
Reviewer 2 Comments |
Author’s response: |
Corrections & Locations |
|
The paper “Development and validation of a parent health- related empowerment scale with low-income parents” reports the development of a 21-item scale to measure parental empowerment through awareness, relationships, and resources (PEARR). |
Correct. Thanks for your assessment. |
N/A |
|
Please include the following information. |
Thank you for your suggestion. We now clarify this information in text as requested. |
Pg. 3, Ln. 117-125
“Item development. The |
|
Indicate how many items were part of the initial pool of items, if the initial pool was different from the final scale. |
|
hypothesized psychological empowerment dimensions in the current study were informed by prior formative research [8]. |
|
|
|
During the formative |
|
|
|
research stage, a |
|
|
|
preliminary pool of 70 |
|
|
|
items spanning a wide |
|
|
|
range of empowerment |
|
|
|
concepts (e.g., collaborative |
|
|
|
competence, bridging social |
|
|
|
divides, facilitating others, |
|
|
|
relational empowerment, |
|
|
|
mobilizing networks, |
|
|
|
critical consciousness) was |
|
|
|
developed . Following an |
|
|
|
extensive review of the |
|
|
|
literature, the authors |
|
|
|
elected to prioritize the |
|
|
|
constructs of critical |
|
|
|
consciousness, relational |
|
|
|
and resource empowerment |
|
|
|
[9, 11, 12, 13, 14, 15, 39, 47]. |
|
|
|
Using an expert agreement |
|
|
|
process, four of the authors |
|
|
|
(RF, JB, KD, JJ) developed a |
|
|
|
preliminary version of the |
|
|
|
scale with 26 items. |
|
|
|
Following pilot testing (described below) 5 items were removed resulting in a final scale with 21 items. ” |
|
Indicate how many items were developed and how many were took from previous scales. |
See the revised section under “Item development”. |
Pg. 3, Ln. 117-125
See above and in text. |
|
How the process of construct validity was conducted? e.g., expert agreement. |
Thank you. We expand on this section of the manuscript to clarify. |
Pg. 3, Ln. 117-125
See above and in text. |
|
Include details about PCA and EFA, e.g., rotation. |
Thank you for pointing this out. Principal component factors within the PCA analysis informed the subsequent EFA step. After EFA was performed, factor loads were rotated to get a clearer pattern (i.e., final factor solution). We add a sentence under statistical analyses and in the results to clarify this step. |
Pg. 4, Ln. 360-361
“Finally, factor loadings were rotated to get a clearer pattern of the items denoting each hypothesized parental empowerment (i.e., final factor solution).” |
|
|
|
Pg. 4-5, Ln. 377-379 |
|
|
|
“Stage 1: Three component factors were identified in the PCA, with eigenvalues of 8.25, 2.75, 2.12. Subsequently, results from the EFA yielded a final three- factor solution with eigenvalues of 5.12, 4.81, 3.19.” |
|
Report the range and distribution of items responses. |
Thank you. We now report on items responses in the manuscript. Sorry we missed this. |
Pg. 4, Ln. 335-336
“All response options for each item ranged from strongly disagree (1) to strongly agree (4) on a 4- point scale.” |
|
Indicate why a PCA was conducted instead of other procedures/criteria to estimate the initial number of factors. |
Thank you. We present this information on page 4, line 156- 160. PCA was proposed at early stages as an item reduction technique. We agree that other procedures would potentially be better suited, but it met the criteria for our overall goal, which it was to examine the scale’s preliminary factor structure and to determine the number of factors to retain prior to conducting EFA/CFA. We are happy to follow-up if you have additional suggestions. |
N/A |
|
Report chi-scare and degree of freedom for the 3-factor model. |
Thank you. It is now reported. |
Pg. 5, Ln. 413-416
“The final factor structure, factor loadings, and model fit statistics for each parental empowerment dimension of the PEARR can be found in Table 3. Fit indices met minimum criteria (X2 [147, N = 337] = 373.78, p < 0.01; CFI = .95; RMSEA = .05 (.05, .06); SRMR = .05).”
Table 3
“Model fit statistics: X2 [147, N = 337] = 373.78, p < 0.01; CFI = .95; RMSEA = .05 (.05, .06); SRMR = .05.” |
|
Report the scale’ instructions |
Thank you. Scale instructions are now included. |
Pg. 4, Ln. 334-335
“Across all sub-scales, participants were instructed to report how much they agreed or |
|
|
|
disagreed with the statements in each scale.” |
|
Report ordinal omega if the Likert scale has less than 10 points. |
As it is our understanding, ordinal omega is an alternative internal consistency test (compared to the Cronbach’s Alpha) for likert-scale responses. We report the alpha coefficients as it is meant to satisfy the same assumptions. Lastly, when doing a search on both techniques within this journal, ordinal omega does not emerge as prevalently as Cronbach’s alpha. |
N/A |
|
Evidence concurrent/discriminant validity. How this scale is correlated with other measures. |
Thank you for such important advice. We now add evidence of concurrent validity in text, specifically, we assess the correlation between health-related parenting empowerment with a validated scale of parenting practices[46]. |
Pg. 5, Ln. 417-420
“Lastly, the PEARR scale showed modest concurrent validity in relation to obesity-related parenting practices focused on physical activity (r=.25-.29) and sleep (r=.25-.37), but not food parenting (r=.02- .16). Summary statistics for each item in the final factor structure for each parental empowerment dimension can be found in Table 4.” |
|
Please clarify
The correlation coefficient between the 3-factor was obtained using each scale item's added scores or was obtained directly from the CFA model. |
The former, it is an average of the total number of items in each sub- scale. |
N/A |
|
The response to the items is a frequency/agreement Likert scale? How many points the scale has? |
Yes, it is an agreement scale. It has 4 points. |
N/A |
|
Were the analyses conducted considering the data as ordinal data? Were the polychromic matrix of correlation computed? Please indicate the algorithm used to obtain the factors for CFA and EFA. |
Thank you for making this observation. Please see included the range of the coefficients in the polychoric correlation matrix across sub-scales. No algorithms were used in CFA/EFA. |
Pg. 4, Ln. 353-354
“Using the full sample, polychoric correlation was used to assess associations across indicators in each sub- scale.”
Pg. 4, Ln. 376-377
“Results from the polychoric correlation matrix indicated items correlated significantly within each hypothesized psychological empowerment dimension (r=.38-.87).” |
|
Items 8 and 16 (I ask a Head Start teacher or staff for help or advice; Head Start influences children’s health) are specific to an intervention or program. Please discuss the role of these items in a study outside of the Head Start program. The items should be changed? Should be eliminated if the scale is administered in a different program? |
Thank you for the suggestion. We are now adding a statement in the discussion about potentially adapting such items in other contexts. |
Pg. 6, Ln. 457-460
“Indicators referencing Head Start could be changed to fit scope of a different program and administered as most relevant to the study context, though in such a case the research team recommends assessing the sensitivity of the adapted items with remaining items in the scale.” |
|
How the items from the scale “Critical Awareness of Socio- Ecological Influences of |
That’s a great question. That is an aspect that we will consider in future studies. In this study, the |
N/A |
|
Health” can be differentiated from beliefs or cultural biases, for example, the item “The things children see on television or in electronic games influence their health.” |
item is not meant to be perceived as positive or negative (if you will), rather a general statement on various socio-ecological influences that respondents may agree or disagree without implying directionality on its effect on health. |
|
|
Please discuss in the limitation section
The impact of the gender bias of the sample |
Thank you. We add a statement on the impact of gender on sampling. |
Pg. 6, Ln. 471-472
“For instance, addressing the imbalance of mothers and fathers may minimize bias from the majority representing the parent sample.” |
Reviewer 3 Report
The PEARR is a very promising instrument and I am already thinking of a few studies I've conducted where it would have been helpful to use the PEARR.
The manuscript is fine to publish as it is, with the exception that an 'R' is missing from PEARR in the abstract and instructions to participants are included in the Materials and Methods section. I wondered if there was an indication of what was meant by 'other parents'. Potentially it could mean a sibling with children or the participant's own parents, but I wondered if it was intended to mean parents outside the family. Was that clear to participants or was it open to their interpretation?
I would like to suggest some additions that would be helpful, but I don't see these as essential.
Include the Chinese and Spanish versions as supplementary files.
Include the number of parents who completed the Chinese and Spanish versions.
It looks as though the numbers completing in Chinese are too small to analyse separately, but I wonder if it is possible to provide Mean/SDs for the different languages and/or different ethnicities. I would predict that the mean scores will be lower for the Chinese parents - assuming they have interpreted 'other parents' as meaning parents who are not within their family.
Author Response
|
Reviewer 3 Comments |
Author’s response: |
Corrections & Locations |
|
The PEARR is a very promising instrument and I am already thinking of a few studies I've conducted where it would have been helpful to use the PEARR. |
Thank you for your review. We are glad to hear your feedback. |
N/A |
|
The manuscript is fine to publish as it is, with the exception that an 'R' is missing from PEARR in the abstract and instructions to participants are included in the Materials and Methods section. I wondered if there was an indication of what was meant by 'other parents'. Potentially it could mean a sibling with children or the participant's own parents, but I wondered if it was intended to mean parents outside the family. Was that clear to participants or was it open to their interpretation? |
Thank you. We made changes accordingly. “Other parents” was used to gather information about parents’ engagement with other Head Start parents. |
See abstract. |
|
I would like to suggest some additions that would be helpful, but I don't see these as essential.
Include the Chinese and Spanish versions as supplementary files. |
Thank you. We know provide these versions as supplemental files. |
See supplemental files. |
|
Include the number of parents who completed the Chinese and Spanish versions. |
Thank you. We added these numbers. |
Pg. 3, Ln. 109-111
“Upon providing informed consent, parents completed the survey, in Chinese (n=15), English (n=611), or Spanish (n=181).” |
|
It looks as though the numbers completing in Chinese are too small to analyse separately, but I wonder if it is possible to provide Mean/SDs for the different languages and/or different ethnicities. I would predict that the mean scores will be lower for the Chinese parents - assuming they have interpreted 'other parents' as meaning parents who are not within their family. |
As mentioned, we are preparing a second measurement paper which will include invariance by parent gender and language of administration. Mean scores by language will be included in this follow paper. . |
N/A |